# Management of Genetic Variation in the Gamete Bank of the Endangered Lake Minnow *Eupallasella percnurus*, Using Genassemblage 2.2 Software

**DOI:** 10.3390/ani12233329

**Published:** 2022-11-28

**Authors:** Dariusz Kaczmarczyk, Jacek Wolnicki

**Affiliations:** The Stanisław Sakowicz Inland Fisheries Institute in Olsztyn, Oczapowskiego 10, 10-719 Olsztyn, Poland

**Keywords:** fish, sperm, cryopreservation, conservation biodiversity, bioinformatic tool, endangered fish species, Genassemblage 2.2

## Abstract

**Simple Summary:**

Maintaining a genetic diversity in populations of endangered species is an important part of conservation biology and its programs focused on maintaining biodiversity. Gene banks are used to safeguard the genetic variability of populations. The management of genetic resources deposited in gene banks requires knowledge of the genetic profiles of the gamete donors and bioinformatics tools to process this information. In this work, we show how to use Genassemblage 2.2 software in managing the genetic variation deposited in a bank of cryopreserved semen. Our results showed that if we combine genetic profiles of gamete donors and bioinformatic techniques, we can identify a small group of semen samples that are enough to transfer all alleles detected across an entire set of samples. Consequently, we recommend the Genassemblage 2.2 software as a convenient tool in the management of genetic variation deposited in gamete banks.

**Abstract:**

The management of genetic resources deposited in gene banks requires knowledge of the genetic profiles of the gamete donors and bioinformatics tools to process this information. In this work, we show how to use Genassemblage 2.2 software in managing the genetic variation deposited in a bank of cryopreserved semen. Our demonstration was based on the leuciscid fish species, lake minnow *Eupallasella percnurus,* which is designated as endangered in Poland. The semen samples (n = 192) were taken from four Polish lake minnow populations and frozen in the gene bank. Fin clips were taken and DNA extracted. Across 13 investigated microsatellite loci, 21–53 alleles were identified in each population and 66 in the entire group of samples. The module “Management of genetic variation in gamete bank” of Genassemblage 2.2 software was used to find the set of samples that will preserve 100% of the detected allelic diversity in the next generation. Our results showed that a small group of 4–19 semen samples was enough to transfer all alleles detected across this set of samples. We, therefore, recommend Genassemblage 2.2 as a convenient tool for the detection of genetic differences between donors, the construction of optimal sets of samples for conservation of genetic variation, and for managing genetic variation deposited in gamete banks. Consequently, it can be used in breeding human-dependent populations and to optimize the use of genetic diversity in samples in the gamete banks. It can be especially useful for conserving populations of species characterized by low genetic variation, such as the lake minnow.

## 1. Introduction

Maintaining a genetic diversity in populations of endangered species is an important part of conservation biology and its programs focused on maintaining biodiversity. It can be performed using techniques of gene banking, which are used in conservation of populations endangered because of anthropogenic pressure or the deterioration of environmental conditions [1,2,3]. The gene bank is a fragment of allelic diversity specific to a given population that has been conserved by using various techniques. These include cryopreservation of gametes, as well as keeping male and/or female stock [4,5,6,7]. Gene banks are used to safeguard the genetic variability of populations [8] and prevent the consequences of catastrophic events leading to extinction. Today, those techniques are applied for both endangered species and economically valuable breeds and breeding lines [1,9,10].

The potential usefulness of gene banks in the conservation of genetic diversity depends on the genetic differences between the individuals from which the gametes are collected [6] and on tools for managing the genetic resources deposited in these banks [10]. It is often observed that genetic diversity is low in endangered populations [11]. Moreover, the individuals used to construct a gene bank can be closely related. Consequently, gene banks may contain samples or individuals that are genetically very similar to each other and using them or their gametes may increase the inbreeding coefficient in offspring generations. In the breeding of animal species, it is important to reduce the probability of a decrease in genetic variation resulting from inbreeding or crossing genetically similar individuals [12]. Therefore, the use of those genetic resources deposited in gene banks without taking into account the genetic differences between the donors is not recommended as it could cause a strong founder effect or inbreeding event and, in consequence, a population without enough genetic variation to survive. It is particularly important to take these effects into account when dealing with populations that are characterized by low levels of genetic diversity and those at a considerable risk of extinction.

The freshwater native fish species, lake minnow *Eupallasella percnurus* (Pallas, 1814) a member of Leuciscidae family, has been used as an example for the use of Genassemblage 2.2 (Poland) software in management of gamete banks. In Poland, this inhabitant of vanishing, small and shallow habitats is considered to be endangered facing extinction, and therefore, is strictly protected by law with the formal requirement of active protection [13]. Active protection measures that include fish translocations to establish new populations, revitalization of their habitats, management of genetic variation and gamete banking are considered necessary for maintaining this species in Poland [13,14]. The Polish populations of this species are known to be characterized by low levels of genetic variation [15]. 

Management of genetic variation in gene banks can be performed if the genetic characteristics of the gamete donors are known and bioinformatic tools are used. Based on molecular markers, e.g., microsatellite DNA, single nucleotide polymorphisms (SNPs), etc., it should be possible to identify genetic differences among males and females used for breeding [16]. Considerable differences between alleles at microsatellite loci suggest that individuals are not closely related to each other [17,18]. Consequently, to identify the most genetically different individuals or gametes, the polymorphism of microsatellite DNA can serve as the basis for genetic profiles of individuals that are included in male or female stocks or that have served as gamete donors [14]. When identifying these individuals, it is advantageous to supplement gene banking techniques with a bioinformatic method designed to speed up the process of analyzing genetic differences and selecting the optimal individuals for breeding. This enables the most valuable sperm samples to be chosen for conserving genetic diversity. Moreover, identification of genetic differences can enable transfer of the maximum possible amount of genetic variation detected across the investigated loci to the offspring [12].

To the authors’ knowledge, no bioinformatic tools that can be used to manage genetic variation deposited in gene banks have been developed so far. Because of this, we developed such a tool and added it to the Genassemblage 2.2 software as a new module, “Management of genetic variation in gamete bank”.

Genassemblage 2.2 is Windows-based software that constitutes a further development of Genassemblage 1.0 [19], including an improved modular interface and expanded functionality. Its installer and a detailed user guide can be downloaded free-of-charge from the author’s website http://pracownicy.uwm.edu.pl/d.kaczmarczyk/main_page.htm (accessed on 4 November 2022). The new module analyzes allelic diversity detected across all loci included in the genetic profiles of males whose sperm samples are stored in a gamete bank. It calculates the overall number of alleles across all loci and indicates which samples should be used to obtain a target percentage of allelic diversity while using as few samples as possible.

Here, we present an example of how Genassemblage 2.2 can be used to manage genetic variation resources deposited in a bank of cryopreserved semen. As an example we used lake minnow males from four Polish populations that had donated semen to that bank.

## 2. Materials and Methods

The investigated lake minnow populations were the following: Mikołajki Pomorskie (MP) (53°50′19″; 19°10′45″), Drozdowo (DR) (54°12′52″; 18°08′21″), Zielonka (ZI) (52°17′46″; 21°08′29″) and Bełcząc (BE) (51°40′29″; 22°35′10″). The semen was taken in 2015 from 192 males (n = 48 fish per population). The fish were captured using baited traps [13] and transported to the laboratory in polyethylene bags filled with water and oxygen. In the laboratory they were maintained in a flow-through tank within water at 15 °C. Approximately 24 h after being captured, the fish were anesthetized by immersion in an aqueous solution of 2-phenoxyethanol at 0.4 g/L. Then, semen samples were taken from each male and cryopreserved according to the procedure of Dietrich et al. [2]. A small fragment of the pelvic fin was taken from each sperm donor for genetic analyses. The fin samples were dried and stored as described by [20]. Shortly (2–3 days) afterwards, all fish were returned to their native water body.

DNA was extracted from all fin samples using a Genomic Mini AX Tissue SPIN DNA Extraction and Purification Kit (A&A Biotechnology, Gdańsk, Poland). The extraction procedure was performed following the manufacturer’s recommendations. The integrity of the DNA samples was visually inspected after electrophoresis in 1.5% agarose gel stained with ethidium bromide. DNA yields were quantified by spectrophotometric analysis. The 13 microsatellite markers known to be polymorphic in this species [20] were used to prepare the genetic profiles of the fish. PCR primers were designed for these microsatellites: *Ca3*, *Ca4* and *Ca12* [21]; *Z9878*, *Z10362* and *Z13419* [22] (primer sequences were taken from the National Center for Biotechnology Information (NCBI) https://www.ncbi.nlm.nih.gov (access data 4 November 2022); and *Eupe1*, *Eupe2*, *Eupe4*, *Eupe5*, *Eupe6*, *Eupe7* and *Eupe9*, which were amplified using primer sequences described by Kaczmarczyk and Gadomski [23] and deposited in the NCBI database. The primer sequences, repeat motifs and accession numbers and details of PCR amplification are given in [20].

The forward primer of each primer pair was the 5′-end labeled with fluorescent dyes (6FAM, VIC, NED, PET). The lengths of the amplified DNA fragments were determined using an Applied Biosystems 3130 Genetic Analyzer against GS400LIZ size standards. Allele determination was performed using GeneMapper 3.0 software (Applied Biosystems, Waltham, MA, USA) according to the manufacturer’s recommendations.

### Genassemblage Setup

Genassemblage 2.2 software, available at http://pracownicy.uwm.edu.pl/d.kaczmarczyk/genasemblage2_2/genassemblage_software2_2.htm (access data 4 November 2022) was used with the module “Management of genetic variation in gamete bank”. The example of input file (Figure 1) was used for the calculations described below. Each of them contained genotypes of 48 males. To perform interpopulation analysis, an input file including genotypes of all 192 investigated males was constructed and named as (ALL).

The Genassemblage 2.2 software was used to identify optimal set of samples on two levels: (1) within each population, (2) in a group containing all four populations and individuals investigated in this study. As a measure of the allelic diversity, a number of alleles detected across all samples and loci were chosen with the software. To obtain the highest possible transfer of allelic diversity to the progeny, the value for the minimum level of genetic variation was set at 100%. The software identified in the input file the samples that contribute most to allelic diversity and enable transfer of a target percentage to progeny by using as few samples as possible. Consequently, all alleles identified in this sample group should be included in the genotypes of sperm donors chosen by Genassemblage 2.2. The details of how to perform this analysis are given in the user manual that can be downloaded from the author’s website, provided above.

## 3. Results

Results of calculations in the output file can be opened by an MS Excel (Microsoft, USA) spreadsheet. This file contains such data as: (1) genotypes of males from the input file; (2) variation across all samples included in the input file (list of alleles at investigated loci); (3) list of samples chosen by the software as optimal for transfer of a target percentage of genetic variation detected across all samples and their genetic profiles; (4) list of alleles at investigated loci that will be transferred by set of samples chosen by Genassemblage 2.2; (5) summary analyses, including number of samples in the input file, number of samples chosen by the software and actual percentage of allelic variation of the entire set that will be transferred. The allelic diversity described in number of alleles is given both in the Genassemblage interface and in the output file. Moreover, the alleles at each locus and across all samples and loci are shown in the output file (Figure 2). An example of the result file for Zielonka (ZI) population is given in Figure 3.

The Genassemblage 2.2 identified 21, 27, 29, 53 alleles across all loci (nA) in the samples from Mikołajki Pomorskie (MP), Drozdowo (DR), Zielonka (ZI) and Bełcząc (BE), respectively (Table 1). In the file containing all samples, 66 alleles were identified. The number of alleles per investigated individual (nA/N) ranged from 0.44 to 1.10 among populations. The lowest value of this indicator was in the group of all samples.

The example of the result file calculated for the Zielonka (ZI) population (Figure 3) shows the genetic profiles of six samples. They enable transfer of 100% of allelic diversity, which are given below the line “selected sample” and the heading “sample”. They are the following: Ziel07, Ziel22, Ziel26, Ziel25, Ziel38 and Ziel48. The subsequent list provides alleles of a given locus, as well as across loci in a group of selected samples.

In the case of other populations, a transfer of 100% of the allelic diversity (N’) is possible if we use 4, 5 and 13 samples from DR, MP and BE populations, respectively. The names of the selected samples from BE, MP and DR populations and the additional information described above are given in the Appendix A. In the group of all investigated samples (ALL), 19 out of 192 are needed to transfer the entire allelic diversity detected in this study. The names of those samples are given in the Appendix A. Summarizing, a transfer of all alleles detected in a given population (N’/N) was possible by using from 8.33% (DR population) to 27.08% (BE population) of samples stored in the gamete bank (Table 1).

## 4. Discussion

The Genassemblage 2.2 software was designed to help in identification of a group of samples deposited in a gamete bank that are most valuable for conservation of the genetic variation of a species. The software can be used for: (1) detecting genetic differences between gamete donors, (2) assembling the samples from the gamete bank in sets that are optimal for transferring allelic diversity to the progeny, (3) management of resources deposited in the gamete bank by maximizing the effect of using a limited number of samples, (4) reduction of the likelihood of using sperm samples from closely related individuals and inbreeding in the next generations. In this paper, we have evidenced that it is not necessary to use all samples from the gamete bank to transfer the entire allelic diversity. In all investigated populations, as well as in the dataset containing all investigated samples, the same effect can be obtained if a small group of highly diverse samples were used. The Genassemblage 2.2 can detect and pinpoint them. This tool can be used in this way to obtain the most diverse progeny and to optimize the use of the available genetic diversity. The software is particularly useful in the case of populations of fish species of low genetic variation, as in our example, lake minnow, when only a small number of individuals differ from the others to a relatively large extent.

Our results give evidence to the fact that Genassemblage 2.2 can serve as a convenient tool for detecting genetic differences between gamete donors and assembling the samples deposited in a gamete bank in optimal sets for conservation of the genetic variation. The Polish lake minnow populations, including those investigated in this study, demonstrate a considerable decrease in the genetic variation resulting from both founder and bottleneck effects and strong genetic drift and inbreeding. Likewise, their low allelic diversity is a well-known feature of this species in its entire range of the occurrence in Poland [20]. 

The small genetic differences recorded in the Polish lake minnow populations indicate close relationships between the fish belonging to them, so it is particularly important to select samples that genetically differ to the largest extent [12]. The lake minnow is a good example of a species where the knowledge of the genetic characteristics of its individuals makes it possible to eliminate genetically similar fish from the conservation programs including their breeding [15].

It should be noted that in this study, lake minnow populations and the number of analyzed markers were chosen for illustrative purposes. It is obvious that for better detection of the genetic differences between lake minnow individuals, more markers should be applied than we used in our work [24]. The use of more markers can increase the number of alleles detected across all investigated markers and increase the size and change the composition of the set indicated as optimal by Genassemblage 2.2. For practical purposes, it would be possible to considerably increase the number of markers because the software does not limit the number of the analyzed markers.

In this work, we have demonstrated that Genassemblage 2.2 software can operate with both disomic and tetrasomic loci such as (*Ca4*). Moreover, the application of Genassemblage 2.2 is not limited to microsatellite markers and can be used with a dataset including more than 100 samples. Other markers that inherit according to Mendelian Laws (such as SNP markers) can be used if numbers are used to name their alleles. This feature makes the software appropriate for conservation of disomic, partially tetrasomic and fully tetrasomic organisms and enables the use of various kinds of genetic data. It is important to note that the set of individuals or their gametes that are analyzed with the module “Management of genetic variation in gamete bank” can also be used to assemble breeding pairs or sets of pairs using other modules included in Genassemblage 2.2, i.e., “Select the best breeding pair and Select individuals for group spawning” [15]. Consequently, this software can be an efficient tool for conservation of genetic diversity in human-dependent populations [12,16,25], including many fish species that are at serious risk of extinction and require active protection measures [26,27].

The potential use of Genassemblage 2.2 is not only limited to conservation biology but can be helpful in specialized aquaculture tasks such as production of all female stock. This software can be applied to help in the selection of optimal sets of the neomales (sex-reversed females) as a source of semen for breeding. In most cases, the neomales lack spermatic ducts, and therefore, fish must be killed in orFder to obtain semen [28]. The potential usefulness of the Genassemblage software is to help obtain genetically diverse progeny, as well as reducing the number of neomales that need to be killed for breeding. In those applications, the Genassemblage software is beneficial in aspects of genetic variation and saving the neomales stock for breeding in the next seasons.

## 5. Conclusions

In this work, we have demonstrated that Genassemblage 2.2 can optimize the use of the genetic diversity deposited in gamete bank. Consequently, this software can be an efficient tool for conservation of genetic diversity in human-dependent populations.

## Figures and Tables

**Figure 1 animals-12-03329-f001:**
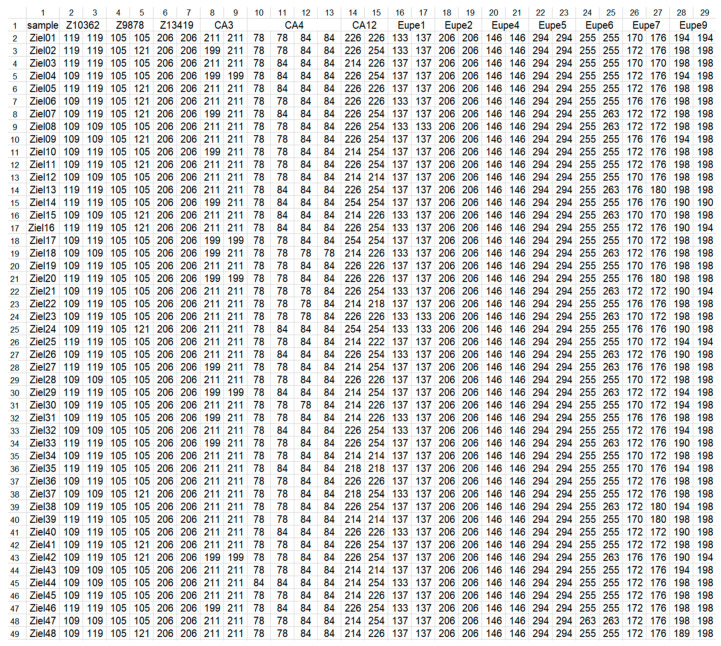
Genetic profiles of lake minnow individuals from Zielonka (ZI) population as an example of input file to Genassemblage 2.2. Column 1, fish ID (samples); columns 2–29, names of microsatellite markers (line 1) and their alleles (lines 2–49).

**Figure 2 animals-12-03329-f002:**
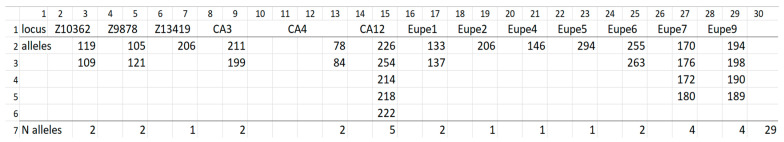
List of alleles detected across all profiles and loci of lake minnow individuals from Zielonka (ZI) population. Line 1, locus name; lines 2–6, alleles of loci enlisted at line 1; line 7, (columns 3–29) number of alleles detected at given locus across of samples and across all loci (column 30).

**Figure 3 animals-12-03329-f003:**
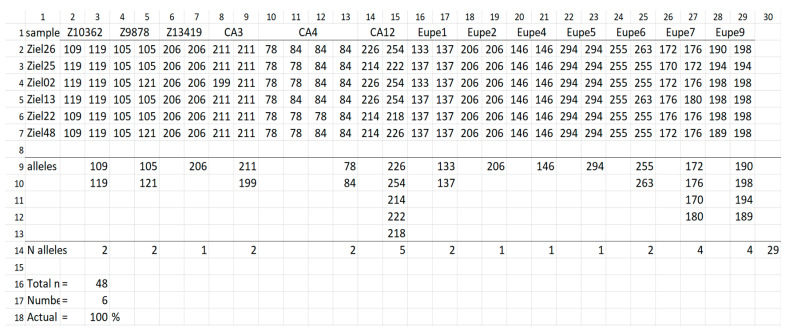
Set of 5 samples that enable transfer to the next generation all alleles detected across genetic profiles of fish from Zielonka population. Line 1, locus name; line 2–7, ID of samples chosen by the software are given in column 1, and their genetic profiles in columns 2–29; line 9–13, headings of lines (column 1, lines 9–18), list of alleles transferred by this set and number of alleles at a given locus and across all loci (line 14 and column 3–30); line 16, total number of samples in input file; line 17, number of samples selected from the input file; line 18, actual percentage of genetic variation detected in the input file that can be transferred to the progeny by using set of samples indicated by software.

**Table 1 animals-12-03329-t001:** Summarized results of the sample selection by Genassemblage 2.2 software. N—number of samples in the input file, nA—number of alleles detected in a given population, nA/N—number of alleles per sample, N’—minimal number of samples required to transfer entire detected allelic diversity to the progeny, N’/N—share (in %) of samples selected by the software in the number of all samples in the input file.

Indicator	Populations				
	Mikołajki Pomorskie (MP)	Drozdowo (DR)	Zielonka (ZI)	Bełcząc (BE)	All Samples (ALL)
N	48	48	48	48	192
nA	21	27	29	53	66
nA/N	0.44	0.56	0.60	1.10	0.34
N’	5	4	6	13	19
N’/N (%)	10.42	8.33	12.50	27.08	9.90

## Data Availability

Not applicable.

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
