# Peer review of "Management of Genetic Variation in the Gamete Bank of the Endangered Lake Minnow Eupallasella percnurus, Using Genassemblage 2.2 Software"

_animals, 2022, doi:10.3390/ani12233329_

Round 1

Reviewer 1 Report

This manuscript presents the application of an updated version of GenAssemblage to the endangered fish Eupallasella percnurus. The author demonstrates how using the software can help in the design of supplemental breeding practices to maximize the genetic diversity of subsequent offspring pools by selecting parents based on microsatellite genotypes.

Overall, the methodology here seemed sound and, for its purpose, the approach can be very useful. As far as impact and originality go though, I do feel the scope and influence of this paper and the gap in knowledge it addresses is somewhat limited. It feels like a nice application of the software and approach which could be applied to very specific cases to some effect. Additionally, the use of this one microsatellite case only limits the impact of the work and its broader applicability. While microsatellite variation is what the author has for work with in this case, this study would have benefited from either a computational study showing the how sample size and number of loci contribute to the efficacy of this program and its estimates of allele number and predicted heterozygosity or a SNP-based analysis that also demonstrated the utility of this approach. Additionally, an approach that can account for the multigenerational/pedigree issues and/or haplotype/diplotype effects on heterozygosity expectations. These kinds of additions would improve the impact of the study. There is some awkward language in the paper and could benefit from proofreading. The supplementals were also incomplete (no video, etc.) and the Figures shown at the end were actually Tables not Figures. The lack of a compelling figure and the tables presenting raw data and results did not aid in the presentation of compelling evidence. Finding a better means of communicating the impact of this work would be advised. Again, a power analysis using computational data would have been nice to see. Overall, I think the manuscript is sound, could impact human-dependent breeding program in fish species such as the one presented here, and could be useful to a certain audience, but does not feel to be written in such a way to gain broader attention to the work.

Author Response

Answer for Review 1.

I would like to thank anonymous Reviewer for reviewing my work.  The comments and suggestions of the reviewer were valuable and very helpful in improving this manuscript.

The list of answers and major changes in this revised version is given below. To make this answer clear and the changes easily identifiable in the revised version, all lines given below are as they appear in the revision of this manuscript. Their line numbers in the originally reviewed version are given in the brackets.

  1. It feels like a nice application of the software and approach which could be applied to very specific cases to some effect.

I thank the reviewer for this comment because it draws my attention to supplementing this manuscript with more cases where my software can be applied. In my opinion this software can be useful in every case when we are looking for set of males as donors of gametes to impregnate eggs. Its use is not limited to conservation biology. For example in can be used for selection neo males (sex-reversed females) for creating all-female stocks for aquaculture purposes. Because neomales in most cases lack spermatic ducts, therefore fish must be killed in order to obtain semen (Nyca et al 2012). A potential use of Geneassemblage software is to help in obtaining genetically diverse progeny as well as reducing the number of neomales that need to be killed for breeding. In those applications the Genassemblage is beneficial in aspects of genetic variation, and saving the neomales stock for breeding in future seasons. This fragment has been added to revised version of manuscript in line 272-280.

  1. Additionally, the use of this one microsatellite case only limits the impact of the work and its broader applicability.

Using the microsatellite DNA as a marker of genetic differences is a demonstration of how this software can be used. But there is no limitation only to microsatellites. The only limitation for using Geneassemblage software is that markers used in assessment of genetic variation must inherit according to mendelian laws. Using SNP markers is possible when we mark their alleles at investigated loci with numeric values. This note is given if section of discussion (line 234-235) and is in the revision at lines 261 and 262.

While microsatellite variation is what the author has for work with in this case, this study would have benefited from either acomputational study showing the how sample size and number of loci contribute to the efficacy of this program and its estimates of allele number and predicted heterozygosity or a SNP-based analysis that also demonstrated the utility of this approach.

This program doesn’t have a limit of used markers and sample size. The number of investigated markers, especially if they are polymorphic, can modify results (line 226-230). Generally, transfer of an entire detected genetic variation in highly genetically divers stock usually require more samples (as a percentage of the investigated group) in the case of more genetically homogenic ones. This may results in differences in composition of sets of samples that are recommended and their number. Generally, precision of the composition set of a recommended sample indicated by Genassemblage 2.2 is as good as molecular technique used in identification of genetic differences between individuals. In real life application, even if we use many molecular markers in detection of genetic differences and detect many alleles across them, this software will enable maximizing the transfer a genetic variation to the next generation while using as small a number of samples as possible.

  1. Additionally, an approach that can account for the multigenerational/pedigree issues and/or haplotype/diplotype effects on heterozygosity expectations.

Thank you for those suggestions.  They are valuable as potential ways of improving Genassemblade software. 

  1. These kinds of additions would improve the impact of the study.

There is some awkward language in the paper and could benefit from proofreading.

Following reviewer’s recommendation a proofreading of this manuscript has been performed by native English speaker.

  1. The supplementals were also incomplete (no video, etc.) and the Figures shown at the end were actually Tables not Figures.

I have prepared a video and it has been added as supplementary material. Name of this file is: How to use Genassemblage 2.2 software in mamagement genetic variation of gamete bank.mp4 This file has been attached to set of files in this revision.

  1. The lack of a compelling figure and the tables presenting raw data and results did not aid in the presentation of compelling evidence.

Examples of genotyping data are given in the input file (Figure1). Moreover the genotypes of males are given in the first section of the output files (supplemental filles A-D). I decided to add this note to the results section in line 164.

  1. Finding a better means of communicating the impact of this work would be advised. Again, a power analysis using computational data would have been nice to see.

I can’t find a suitable analysis to add and increase impact of this work whilst staying within aim of this work.  Maybe this is a good idea for further work and improvements of this software.

Reviewer 2 Report

General comments

After reviewing the manuscript titled “Management of genetic variation in the gamete bank of the endangered lake minnow Eupallasella percnurus, using Genassemblage 2.2 software”, the author describes a software revision. This software is a mature version. I only have a few concerns preventing me from recommending the paper's publication in its current form. I suggest that the author must put the figure in the main text. This can help readers understand the content of the article more efficiently. This is an excellent study and I have no other suggestions for revisions. I recommend that the manuscript will not be officially accepted for publication until editing of the English grammar and phrasing by a native English speaker. In my opinion, this manuscript does not meet the criteria for publication and must therefore be a minor revision.

1. What is the main question addressed by the research?

→ This is a revision of the software, the old version has been published in other scientific journals in the past (e.g. Scientific Reports).

Kaczmarczyk, D., & Wolnicki, J. (2020). Genassemblage 2.0 software facilitates conservation of genetic variation of captively propagated species. Scientific Reports, 10(1), 1-7.
2. Do you consider the topic original or relevant in the field? Does it
address a specific gap in the field?

→ This research is an original subject that complements the field.
3. What does it add to the subject area compared with other published
material?

→ This is an interesting piece of software, the author keeps updating the version.
4. What specific improvements should the authors consider regarding the
methodology? What further controls should be considered?

→ Since this is an updated version of the software, I think this is an acceptable article based on past publications
5. Are the conclusions consistent with the evidence and arguments presented
and do they address the main question posed?

Yes
6. Are the references appropriate?

Yes
7. Please include any additional comments on the tables and figures.

I suggest that the author must put the figure in the main text. This can help readers understand the content of the article more efficiently.

Author Response

Answer for Review 2.

I would like to thank anonymous Reviewer for reviewing my work.  The comments and suggestions of the reviewer were valuable and very helpful in improving this manuscript.

The list of answers and major changes in this revised version is given below. To make this answer clear and the changes easily identifiable in the revised version, all lines given below are as they appear in the revision of this manuscript. Their line numbers in the originally reviewed version are given in the brackets.

  1. What is the main question addressed by the research?

→ This is a revision of the software, the old version has been published in other scientific journals in the past (e.g. Scientific Reports).

Kaczmarczyk, D., & Wolnicki, J. (2020). Genassemblage 2.0 software facilitates conservation of genetic variation of captively propagated species. Scientific Reports, 10(1), 1-7.

I decided to add in this answer for the reviewer  some explanation and supplementation to this point. In those studies we use an improved version of Genassemblage 2.0. The aim of the studies presented here was different than in publication mentioned above. This work presents how Genassemblage 2.2 software can be use in management of genetic resources in gamete bank. The manuscript published in Scientific Reports 2020 presents how this software can be use in assembling individuals in pairs or groups for spawning in order to maintain genetic variation in human dependent population or stock. In summary, in those two works not only was the aim of study different but also the bioinformatic methods implemented and used in Genassemblage software were different.

  1. I recommend that the manuscript will not be officially accepted for publication until editing of the English grammar and phrasing by a native English speaker.

Following Reviewer’s suggestions proofreading of this manuscript has been done by a native English speaker.

  1. I suggest that the author must put the figure in the main text. This can help readers understand the content of the article more efficiently.

Following the Reviewer’s suggestions I have put all figures into the text.